# Does the Back Plate Position Influence Swimming Start Temporal Characteristics?

**DOI:** 10.3390/ijerph19052722

**Published:** 2022-02-26

**Authors:** Daria Rudnik, Marek Rejman, Leandro Machado, Ricardo J. Fernandes, João Paulo Vilas-Boas

**Affiliations:** 1Faculty of Physical Education and Sport, Wrocław University of Health and Sport Sciences, 51-612 Wrocław, Poland; marek.rejman@awf.wroc.pl; 2Centre of Research, Education, Innovation and Intervention in Sport (CIFI2D) and Porto Biomechanics, Laboratory (LABIOMEP-UP), Faculty of Sport, University of Porto, 4099-002 Porto, Portugal; lmachado@fade.up.pt (L.M.); ricfer@fade.up.pt (R.J.F.); jpvb@fade.up.pt (J.P.V.-B.)

**Keywords:** biomechanics, swimming, kick-start, back plate, starting platform, preference effect, sex

## Abstract

The current study examined the back plate position impact on the block phase movement pattern and total starting performance with a distinction for sex. Thirty-eight swimmers performed starts changing the back plate position (preferred position, one position forward and one position backward), with the data being assessed using a 3D dynamometric central and a video camera. In males, the 15 m start time was 0.1 s shorter for the preferred position compared with the backward position (*p* < 0.05). Regardless of the back plate positioning, the swimmers spent a similar time on the block. A more forward position of the back plate postponed the rear foot take-off and consequently reduced the front foot stand duration. A back plate position effect was revealed for those variables with a larger effect size in males whereas in females, a change of about two positions was needed to reach a significance level. Probably due to the specification of physical domains, a greater impact on the changes introduced was noted for males. Therefore, whilst searching for the optimal starting position, adjustments to the back plate placement might affect a greater number of males than females. To reinforce the starting optimization during the training process and its monitoring, the effect of personal preference toward the starting block setting was also considered.

## 1. Introduction

To excel in any sport, it is necessary to optimize all performance components. Specifically, in competitive swimming (an individual and cyclic sport), the main goal is to swim as fast as possible over a set distance. Therefore, a competitive swimming event should be analyzed with regard to its distinct phases, i.e., the starting, swimming, turning and finishing segments [1,2]. Following Federación Internacional de Natación (FINA)-specific rules, the start phase can last up to 15 m from the starting line with swimmers performing actions in both terrestrial and water environments. Thus, to propel themselves from the starting block, swimmers involve their whole body (even if the lower limbs are the main impulse generators [3,4]), with each limb role highly dependent on its relative positioning.

It is known that the initial starting position determines not only the overall performance of the start but also the characteristics of its elements [2,5,6]. Particularly in the kick-start, the lower limbs are placed in a staggered position [2] with the rear lower limbs having a significant importance in producing horizontal velocity and the front lower limbs contributing decisively to vertical velocity production and body weight support [7,8,9,10]. Moreover, each lower limb plays a different role in the inverted pendulum approach as the model of a kick-start [7]. Thus, each lower limb contribution and placement is reflected in the take-off velocity component and modifies the take-off angle, affecting the profile of the subsequent starting actions.

The currently used starting block is approved by the FINA and equipped with an adjustable incline part (the back plate) that is fixed to the main deck in five locations and inclined by 30°. Placing the rear foot upon a back plate has made the kick-start more advantageous compared with the previously used track start technique [10] and changing its positioning might influence both the lower limb joint angles and the position of the initial center of mass [8]. Therefore, by adjusting the back plate position, a swimmer is able to find an optimal body position considering his/her body dimension. Greater differences between the sexes in the block time as a consequence of the kick plate implementation have already been described [11].

Before the introduction of the back plate, researchers focused on the impact of the placement of the body on the starting platform as well as on the effect of changes to the specific variable values and on the duration of swimming start phases to find a detailed contribution of the given solutions to the starting performance output [2,12]. As the back plate has become more widely implemented and used during competitions, researchers have aimed to expose the importance of lower limb positioning over the starting platform (including the distance between the feet [13]) as well as the impact of the lower limb joint angles on the swimming start performance [14]. Several studies have hypothesized that the starting performance could be affected by only adjusting the back plate [15,16,17]; others have analyzed back plate displacement combined with different initial body positions [14,17,18,19]. However, a few of the available studies were based on block settings incompatible with official standards and included multiple factors acting on the measured output or did not consider sex-specific differences.

It is unquestionable that a starting performance improvement requires more comprehensive research because the back plate positioning is generally selected based on the personal preferences of swimmers [15]. The movement pattern can also be affected by previous practice experience [20]. Thus, swimmers may tend to follow their usual movement patterns, particularly by adjusting their body position to the provided starting block features [21]. Despite the availability of findings exposing positions other than a preferential one as equal to or even more advantageous [18], the most frequently practiced technique has been very often qualified as the best to be used [22,23]. Therefore, the decisions involved in the swimming start training process optimization should be supported by multidirectional research and findings implied by reliable sources of information. As there is no sufficient information regarding the effects of different back plate positions, it should be examined how back plate position adjustments could influence the starting performance of male and female swimmers.

The aim of the current study was to evaluate the ventral start performance using different back plate positions and to observe the eventual movement pattern adaptations of swimmers associated with those changes. To verify and quantify the temporal differences between the trials (incorporating preferred back plate position, one position forward and one position backward from the preferred back plate position), a particular emphasis was put on block phase analyses. The effect of the individual preferences of back plate positioning was also considered and the results were analyzed, distinguishing male and female swimmers.

## 2. Materials and Methods

### 2.1. Participants

Thirty-eight international-level swimmers with a best competitive performance ≥ FINA 750 points voluntarily participated in the current study. This group was composed of 19 females and 19 males with 16.6 ± 2.2 vs. 20.8 ± 4.2 years of age, 169.7 ± 4 vs. 179.1 ± 6.4 cm of body height and 59.9 ± 4.5 vs. 73.4 ± 9.0 kg of body mass, respectively. Before the testing sessions, the swimmers and their coaches were informed about the study purpose and the experimental procedures. During the data acquisition, all participants were healthy, without any injuries and rested from any fatiguing exercises. The research protocol, consistent with the Declaration of Helsinki, was approved by the local Ethics Committee. All swimmers—or if under 18 years old, their legal guardians—signed written informed consent forms.

### 2.2. Procedure

Firstly, the body height, body mass, age, competitive experience and individually preferred position of the back plate were assessed. A warm-up based on a standard pre-race routine was then implemented by the participants. The swimmers were already accustomed to the kick-start swimming technique. Each subject performed three series of three repetitions in each of the kick-start conditions and, based on their previous experience, the individually preferred position of the back plate was selected. To obtain a full set of back plate positions to be considered, positions other than the preferred one were then revealed. These comprised one position forward and one position backward from preferred back plate position. Finally, with the collected trial options, each swimmer had all the trials arranged in a randomized order. It is worth underlining that the participants were free to choose their preferred movement pattern whilst starting.

The starting procedure complied with the FINA rules and was organized under simulated race conditions to ensure the best possible starting performance. After starting, the swimmers were requested to swim a front crawl for at least 20 m from the starting line to ensure representative values of the 15 m start time. The participants were asked to accomplish each repetition in the shortest possible time. To recover from fatigue, at least three minutes of a break interval was ensured between the trials. The starting signal (acoustic and optical) was given to the swimmers with a dedicated device (Onda TTL, 0–5 V, Porto, Portugal). In addition, it also allowed us to simultaneously initiate and synchronize the video recordings and dynamometrical data collection. The testing sessions were performed in an indoor 25 m swimming pool and the FINA regulations regarding the facilities were followed.

### 2.3. Measurements

One surface video camera (GoPro Hero 4, GoPro, San Mateo, CA, USA), recording at 50 Hz, was used to measure the 15 m start time. The camera was fixed to a tripod (Hama Star 63, Hama Technics SL, Barcelona, Spain at a height of 0.5 m perpendicular to the trajectory of the body of the swimmer during the start. A light-emitting diode (LED; a light connected to a trigger) was used to synchronize the camera with the starting signal. A 3D dynamometric central with a sampling frequency of 2000 Hz (3D-6DoF, corresponding with the starting block OMEGA OSB 14) and Visio software (LabVIEW 2013 System Design Software, SP1 NI^TM^, Austin, TX, USA) were used to accurately measure the temporal variables of the block phase of the ventral starts [4,24]. The description of the measured variables is presented in Table 1; these were selected based on the literature [2,25].

### 2.4. Data Processing

The best trial was selected for a further analysis on the basis of the 15 m start time. It was taken for granted that the shorter the 15 m time, the better the starting performance was. Firstly, key biomechanical variables were selected then their values were measured from the collected data with the use of dedicated software. The video recordings were treated with DaVinci Resolve software (Blackmagic Design Ltd., Fremont, CA, USA); the first frame with a visible LED light was used to determine the starting signal for a given trial. A processing routine created in MATLAB R2016a software (MathWorks Inc., Natick, MA, USA) was employed to derive the temporal characteristics of the block sub-phases on the basis of the data collected with the 3D dynamometric central.

### 2.5. Statistical Analysis

Before examining our research questions with statistical tests, the collected data were evaluated using parametric test assumptions. To describe the group with representative values of the obtained results, the means and standard deviations were computed for all the variables. A repeated analysis of variance was run to compare the variables extracted from the repeated observations of the three different swimming start variants defined by the position of the back plate. In cases revealed as significant through the ANOVA, a Duncan post-hoc test was used to verify the significance for three dependent pairs of measurements. Several differences among the variable values were brought to our attention. Those results, obtained from a one-way ANOVA, did not reach a significance level (*p* > 0.05); thus, the requirement for further testing with a post-hoc test was not achieved. Therefore, a further aim was to consider the possible consequences of the specific changes applied to the back plate position for a given direction in greater depth. A *t*-test for repeated measures was performed for the variables measured in every two pairs from the block configurations tested. To augment the significance test results, the effect size was reported [26]. The statistical procedures were conducted using Statistica 13.1 software (StatSoft Tulsa, OK, USA) with the level of statistical significance established at α = 0.05.

## 3. Results

The descriptive characteristics of the temporal variables measured for the three tested conditions complemented with the differences between the trials exposed through the statistical procedures and their size are presented in Table 2 and Table 3 for the female and male participants, respectively. In the female group, the average value of the 15 m time measured for all conditions (7.31 s) was higher than in the male group (6.39 s). Following our expectations, the male participants needed less time to cover the 15 m distance after the start than their female counterparts (*p* < 0.000) in all of the variants considered. For both groups tested, the back plate position mostly influenced the foot contact times (rear foot take-off and front foot stand). A comparatively high value of the standard deviation in the hands take-off time suggested a high intragroup variability as the vast majority of the participants managed the movement of their upper limbs in an individually specified (and probably preferred) pattern.

In the male group, the 15 m start time was 0.103 s shorter for the preferred back plate position compared with the backward one (*p* = 0.023). A back plate position effect was also noted for the rear foot take-off and front foot stand times (*p* < 0.001). The time that elapsed from the starting signal to the rear foot take-off decreased when the back plate position was changed from the front to further toward the back. Simultaneously, an increase in the front foot contact time was observed. Here, the variables describing the temporal organization regarding the lower limb push-off time distribution showed the tradeoff between the extension of the rear foot take-off and a decrease in the front foot stand.

No back plate position effect was observed for the front foot stand and for the rear foot take-off time (*p* = 0.073 and 0.072, respectively) for the female swimmers The most remarkable effect of lower limb movement organization during the block phase was brought about by shifting the back plate two positions. A change in the back plate position from forward to backward with regard to the preferred back plate position resulted in an extension of the time spent for the front foot stand of 0.013 s and a shortening of the rear foot take-off time of 0.043 s. Therefore, a similar tendency was observed in the female group but without a statistical significance in most cases.

## 4. Discussion

The purpose of this study was to answer the question of how adjustments to the starting block structure would influence the start characteristics depending on the sex of the swimmers. This study aimed to assess the swimming start performance with different positions of the starting block back plate and to identify if adaptations would occur in the movement patterns of swimmers in association with those position changes. The effect of the individual preferences of back plate positioning on the overall starting performance was also taken into account.

### 4.1. Overall Starting Performance

In the current study, no back plate positioning effect was noted for the 15 m start time. However, in the males, the 15 m start time was lower for the preferred back plate position than in its backward adjustment (Table 3). Considering the competitive level of the participants and that a small time difference can decide the winner, the advantage in the total starting time of 0.1 s and 0.07 s for males and females, respectively, measured in the current study might be significant for coaches and swimmers. Previously, a 5 m start time for starts incorporating a back plate placement at 0.29 m was longer compared with other conditions (0.44 or 0.59 m) tested [13]. In contrast, the only study that examined the impact of a back plate adjustment with a 15 m start time showed no differences among the tested back plate positions [15]. Unfortunately, the mentioned study did not consider the preferred starting block setup of the participants (the back plate position changes were based on the shin length of the swimmer) and both sexes were combined in one analysis. No effect of the back plate position was noted for a 7.5 m start time [17]. According to those authors, the limited availability of the new starting block during daily practice could be important for the level of success. The movement output could significantly be affected by the experience gained throughout previous practice [20]. The addition of new features requires a sufficient time for swimmers to adjust the neuromuscular properties toward the changed conditions [27]. Therefore, to allow swimmers to become acquainted with a new starting block specification, dedicated technical training with settings compatible with a swimming competition have to be included in daily practice.

### 4.2. Effect of Preference on Back Plate Positioning

In our study, it was found that males obtained a shorter mean 15 m start time using their preferred positioning (Table 3). It has been noted that a starting technique specialized throughout the training of a swimmer ensured a better mastery [28]. The most practiced technique often guarantees the best starting performance [22,23]. In a study evaluating the differences in the preferred back plate position with the consideration of anthropometrical characteristics, only one-third of the participants displayed a decline in the 15 m start time as a result of changes to their preferred position [18]. It has to be highlighted that the movement output is also influenced by the past experience of an athlete as a tendency toward selecting a strategy congruent with the previously mastered one has been exposed [20]. Thus, swimmers searching for comfortable, established and stable circumstances tend to adjust their implemented position to obtain starting conditions possibly similar to their well-known ones [21].

In the current study, all participants were allowed to perform a few practice starts before the data acquisition but no extensive training was provided. The preferred back plate position was defined by each subject on the basis of their previous experience. It has been demonstrated that the higher the level of experience in swimming starts, the better the results of its performance [22,23]. The relationship between experience, preferences of swimmers and their starting performance has been evaluated [5]. The greatest instantaneous horizontal velocity at 5 m presented the highest positive correlation with the preferred starting position (r = 0.53). This reasoning could explain the results obtained in the current study, suggesting the superiority of a start employing preferential starting block settings. In a few cases, the preferred starting position was so well-established that the swimmers tended to control the accommodation of their position in an unpreferred setup of the starting block by lifting or lowering their foot over the inclined back plate [21]. This can also imply relevant biomechanical effects.

As mentioned above, whilst searching for optimal conditions to improve their starting performance, swimmers tend to adjust their setup body position [21]. This might be a consequence of incorporating the unique physical attributes of the subject or rather a habituation effect. The non-preferential technique was found to be less stabilized and described by a higher inter-trial variability [23], which could explain its lower efficacy. It could be a psychological effect arising from the comfort of the swimmer, skill stability or fear of making a mistake. It has been demonstrated that starts other than a preferred one may provide further improvements to the starting performance after extensive practice [22]. Therefore, the adaptation of the swimming start as a result of searching for an optimal movement pattern (technique) on the basis of biomechanical criteria should be recommended for coaches and swimmers.

### 4.3. Adaptation to the Pattern of Movement Organization

The symptoms of adaptation that occurred in the movement patterns of the participants in association with the implemented back plate positioning did not influence the block phase duration (Table 2 and Table 3). Consequently, the swimmers spent a similar amount of time on the starting block (ranging from 0.76–0.77 ± 0.03 s and 0.72–0.73 ± 0.04 s for females and males, respectively). Comparable results with a mean block time of 0.77 ± 0.01 s (*p* = 0.089) for all tested kick plate positions was observed in a previous study [17]. Likewise, in other studies, the block time did not vary depending on the back plate position [14,15] although it was noted that the reaction time decreased when the back plate was placed at a distance equal to the shin length of the swimmers from the front foot but this exerted no effect on the block time (0.69–0.72 s) [15]. Interestingly, a study evaluating a wide number of alternatives to the preferred back plate position did not show any differences in the block time between a high front center of mass position combined with a narrow stance and a low front center of mass position combined with a wide stance [18].

Temporal organization regarding the lower limb contact time with the starting block showed the tradeoff between the extension of the rear foot take-off and a decrease in the front foot stand (Table 2 and Table 3). A decrease, in percentage values, of the time spent only for the front foot contact resulting from the back plate forward position was previously reported [13]. That change affected the acceleration profile of the body of a swimmer [13], which was a consequence of a modification in the distance between the hips of the swimmer and the edge of the back plate. The length of the lower limb muscle tendon units might change in those conditions, which, in turn, might impact on the efficiency of the force production [29]. During the block phase, each lower limb contributes differently to the velocity production [9,10]; impulse is the integral of the force applied during a given time interval. On this basis, when a timing transition between the push-off time of the lower limbs takes place and its segment positioning is slightly changed, a transfer between the magnitudes of the velocity vectors is also be reflected [7]. Consequently, it has been demonstrated that a more backward position of the back plate results in a higher horizontal take-off velocity [13,14,17].

As a consequence of back plate adjustments, the characteristics of each lower limb action might differ. Similarly, in sprint track and field starts, the elongation of the foot position in the starting block allows the generation of greater take-off forces [30,31]. In a sprint start, the priority should be to maximize the anteroposterior bilateral force production rather than the subsequent unilateral force to enhance performance [25]. Furthermore, in a sprint start, lower center of mass projection angles at the end of each sub-phase of the block phase were associated with a better performance [25]. It might, therefore, be difficult to distinguish the response to the change of the back plate position from that of the take-off angle. Both conditions were included in a study searching for consequences brought by different starting block setups on the swimming start [13]. However, they provided promising results regarding the enhancement of the starting performance. Searching for optimal conditions for the efficiency of the musculoskeletal system in the starting position can help reinforce the effect of personal preference in the block features.

A swimming race is a sum of different phases that includes not only free swimming but also other technical elements [1]. However, the start always initiates rivalry. Therefore, depending on the body position, the contribution of each lower limb and its placement can impact on the take-off features and, consequently, consecutive elements of the swimming race. When leaving the block, a swimmer needs to find a proper take-off angle combined with the forward rotation of the body to generate a sufficient angular momentum and make a proper entry into the water [2,32]. In addition, the flight distance (in relation to the body height) has been exposed as positively correlated with the average vertical force exerted during the front foot stand (r = 0.783) [7].

### 4.4. Sex Effect Impact on the Start

The results of this study pointed out a lower mean 15 m start time (0.921 s) obtained by the male swimmers compared with their female counterparts. In the males (with the effect size of η_p_^2^ = 0.14), the longest total start time (15 m) was observed for the trials with a backward back plate position accounting for 101.6% of the shortest 15 m start times obtained with the preferred back plate position. These results were consistent with those achieved in the majority of studies presenting a shorter start time for male than for female swimmers [33,34,35]. As swimming performance depends on many factors [36], physical strength and technical diversity between the sexes were also reported as factors influencing the start [35]. Calculated after McClelland and Weyand [37], the percentage difference in the male and female total start time was 14.4%; for the whole swimming race, that value was much lower (7–11%) [38]. This suggested that the sex effect differences had a greater influence on the starting performance than the total event time. Findings from other sports also show a diversity in the sex-based skill gap, depending on the event. The mean male/female differences across jumping events were greater (17.8 ± 2.7%) than the respective mean differences for running events (11.2 ± 1.4%) [37].

A higher effect size was observed for the males not only for the overall starting performance but also for the temporal profile of the lower limb movement organization during the block phase. A sex effect was exposed for the reaction time when auditory stimuli were provided [39]. The body dimension can influence not only the body position but also the contact time of lower limbs with the starting block. If each lower limb contributes differently to the profile of velocity development [9,10], the presented temporal structure variability may affect the take-off velocity and take-off angle of each sex differently. The change in the back plate position has been shown to influence the block time [16], which, consequently, should modify the ability to generate the take-off forces [29,40]. A higher muscle power leads to an improvement in the start impulse among male swimmers [33,41] and the peak forces produced by females on the block have been shown to be lower compared with those in males [41]. Therefore, various adjustments to the back plate position probably affect males more than females.

## 5. Study Limitations

Notwithstanding the pertinence and originality of the study, a few limitations and future research directions should be addressed. Firstly, the current study explored only three of the five available back plate positions (the preferred position of the swimmers, one above and one below). Yet, the majority of research that evaluated corresponding issues typically focused on the same amount of starting block setups [13,14,15,16,17,21]. The starting block used in our study emulated the OSB 11 and the swimmers chose their own preferred positions (individuality inclusion) based on their previous experience. To ensure that their own bias, habits or psychological effects of the swimmers were not the main factors that influenced the obtained results, an upcoming study should include an extended period of adaptative training with non-preferential variants of back plate positioning. We are also aware that only selected biomechanical variables of the block phase were taken under consideration but the most relevant, in terms of chosen issues, were presented and discussed in the wide context of the findings provided by other studies.

## 6. Conclusions

This study presented the superiority of the preferential back plate position for the ventral start performance of male swimmers compared with the conditions incorporating the backward back plate position. In general, regardless of the back plate positioning, swimmers of both sexes tended to spend a similar time on the starting block. However, a variability among the tested positions was observed with reference to the duration of each lower limb stand time. A more backward back plate position ensured a shortening of the rear foot take-off time and an extension of the front foot stand. Therefore, whilst searching for the optimal conditions for the efficient functioning of the musculoskeletal system during the initial starting position and subsequent block actions, the effect of the subjective preference of the back plate position should be taken into consideration. Moreover, it seemed that the various adjustments to the back plate position affected the males more than the females. Adaptation as a result of searching for an optimal movement pattern of the swimming start based on biomechanical criteria should be recommended for coaches and swimmers. Consequently, the exposure of strengths and weaknesses of the back plate positioning variants provides utility knowledge, which should lead coaches and swimmers in the optimization process to exceed the current performance of the swimmer.

## Figures and Tables

**Table 1 ijerph-19-02722-t001:** Definitions of the variables used for the swimming start analyses.

Variable	Definition
Reaction time (s)	The time interval between the starting signal and the first observable change in the starting block reaction force to time curve as a result of the initial movement of a swimmer
Hands take-off (s)	The time interval between the starting signal and the last contact of the hands with the starting block
Hands take-off: reaction time (s)	The time interval between the starting signal and the last contact of the hands with the starting block, reduced by the reaction time
Rear foot take-off (s)	The time interval between the starting signal and the last contact of the rear foot with the starting block
Rear foot take-off: reaction time (s)	The time interval between the starting signal and the last contact of the rear foot with the starting block, reduced by the reaction time
Front foot stand (s)	The time interval between the last contact of the rear foot with the starting block and the moment when the total vertical force fell to zero
Block time (s)	The time interval between the starting signal and the moment when the total vertical force fell to zero
Movement time (s)	The time interval between the first visible change in the starting block reaction force to the time curve and the instant when the total vertical force fell to zero
15 m time (s)	The time interval between the starting signal and the moment when the head crossed the 15 m mark

**Table 2 ijerph-19-02722-t002:** Mean ± SD of the temporal variables of swimming start performed by female swimmers presented separately for each starting position and complemented with significant results exposed through statistical procedures.

Variable	Back Plate Positions			Effect Size η_p_^2^
	Forward	Preferred	Backward	
15 m start time (s)	7.351 ± 0.32	7.282 ± 0.33	7.306 ± 0.37	0.074
Reaction time (s)	0.165 ± 0.03	0.167 ± 0.03	0.158 ± 0.03	0.081
Hands take-off (s)	0.457 ± 0.09	0.441 ± 0.08	0.449 ± 0.08	0.085
Hands take-off: reaction time (s)	0.287 ± 0.08	0.269 ± 0.06	0.283 ± 0.08	0.105
Rear foot take-off (s)	0.641 ± 0.04 ^b^	0.630 ± 0.03	0.618 ± 0.05 ^f^	0.136
Rear foot take-off: reaction time (s)	0.475 ± 0.04	0.463 ± 0.03	0.460 ± 0.03	0.106
Front foot stand (s)	0.131 ± 0.02 ^b^	0.140 ± 0.02	0.144 ± 0.02 ^f^	0.135
Block time (s)	0.772 ± 0.03	0.769 ± 0.03	0.761 ± 0.05	0.058
Movement time (s)	0.607 ± 0.04	0.602 ± 0.03	0.603 ± 0.04	0.017

^f^ and ^b^: different forward and backward back plate conditions, respectively.

**Table 3 ijerph-19-02722-t003:** Mean ± SD of the temporal variables of swimming start performed by male swimmers presented separately for each starting position and complemented with significant results exposed through statistical procedures.

Variable	Back Plate Positions			Effect Size η_p_^2^
	Forward	Preferred	Backward	
15 m start time (s)	6.411 ± 0.47	6.331 ± 0.55 ^b^	6.434 ± 0.49 ^p^	0.137
Reaction time (s)	0.168 ± 0.04	0.175 ± 0.03	0.171 ± 0.03	0.042
Hands take-off (s)	0.452 ± 0.07	0.463 ± 0.08	0.445 ± 0.07	0.021
Hands take-off: reaction time (s)	0.279 ± 0.07	0.288 ± 0.08	0.276 ± 0.07	0.008
Rear foot take-off (s)	0.609 ± 0.04 ^p, b^	0.615 ± 0.05 ^b, f^	0.589 ± 0.05 ^p, f,^ *	0.545
Rear foot take-off: reaction time (s)	0.448 ± 0.05 ^b^	0.440 ± 0.05 ^b^	0.424 ± 0.04 ^p, f,^ *	0.425
Front foot stand (s)	0.109 ± 0.02 ^p, b^	0.118 ± 0.02 ^b, f^	0.130 ± 0.02 ^p, f,^ *	0.554
Block time (s)	0.718 ± 0.04	0.734 ± 0.05	0.719 ± 0.04	0.061
Movement time (s)	0.557 ± 0.05	0.558 ± 0.04	0.554 ± 0.05	0.032

* Significant back plate position effect at exactly *p* ≤ 0.05. ^p^, ^f^ and ^b^: different preferred, forward and backward back plate conditions, respectively.

## Data Availability

The data that support the findings of this study are available on request from the corresponding author (D.R.). The data are not publicly available due to ethical reasons.

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
