# Peer review of "Does the Back Plate Position Influence Swimming Start Temporal Characteristics?"

_ijerph, 2022, doi:10.3390/ijerph19052722_

Round 1
Reviewer 1 Report
Overall, the paper is good, organised, well written and the content is significant for the field of sports (swimming) in finding new techniques for improving the swimmers’ performance.
Some strengths related to both scientific content and form are:
- Based on previous similar works, the authors identified the unsolved challenges and proposed a study to examine how the adjustments of the back plate position would influence the starting performance of the swimmers of different gender.
- The paper is well-structured and documented.
- An overview on related works, some from recent years, is presented. It contains studies from specific literature about how different methods or techniques were used.
- The authors describe their original contribution related to the topic in discussion, also by conducting the study and analysing the results separately for females and males.
- The research design is well-defined and justified from a scientific point of view. It is also approved by the ethical committee.
- The theoretical background is well explained and properly used.
- The research design contains experiments which are well described, and the results are presented and discussed accordingly.
- The conclusions are clearly defined, and future work is identified, mostly based on the limitations of the study.
- The references are in accordance with the topic of the paper.
Some suggestions are:
- The Abstract should better emphasize that the study will focus not only on the impact of back plate position on the block phase movement pattern, but also for the differences related to male and female swimmers.
- More details are needed to justify why the specific statistical test were applied in the analysis of the results.
- The study has some limitations, but they are already identified by the authors.
- Some small corrections should be made related to the use of English language.
- In some situations, it is proper to use the word “gender” instead of the word “sex” (e.g. sex-effect, sex gap, etc.).
Reviewer 2 Report
Making the indicated modifications, the study is novel and interesting

Reviewer 3 Report
This study investigated the impact of back plate position on block phase movement patterns and the starting performance evaluation for the swimmers. Some revising need to be updated as follows:
The abstract should have more discussion and take-home messages for the audience.
Introduction
The background and idea flow are not quite clear, the logic for the purpose of this research need to be strengthened.
Materials and Methods
Lines 158, missing the period.
Discussion
Lines 230 – 232, should have more discussion of the new starting block during daily practice for the level of success.
